# Low-Dose Radiation Exposure with ^56^MnO_2_ Powder Changes Gene Expressions in the Testes and the Prostate in Rats

**DOI:** 10.3390/ijms21144989

**Published:** 2020-07-15

**Authors:** Nariaki Fujimoto, Gaukhar Amantayeva, Nailya Chaizhunussova, Dariya Shabdarbayeva, Zhaslan Abishev, Bakhyt Ruslanova, Yersin Zhunussov, Almas Azhimkhanov, Kassym Zhumadilov, Aleksey Petukhov, Valeriy Stepanenko, Masaharu Hoshi

**Affiliations:** 1Research Institute for Radiation Biology and Medicine, Hiroshima University, Hiroshima 7340037, Japan; 2Semey Medical University, Semey 071400, Kazakhstan; gauhar2101@mail.ru (G.A.); n.nailya@mail.ru (N.C.); dariya_kz67@mail.ru (D.S.); zhaslan_love@mail.ru (Z.A.); baharuslanova@gmail.com (B.R.); smu@med.mail.kz (Y.Z.); 3National Nuclear Center of the Republic of Kazakhstan, Kurchatov 071100, Kazakhstan; azimhanov@nnc.kz; 4L.N. Gumilyov Eurasian National University, Nur-Sultan 010000, Kazakhstan; zhumadilovk@gmail.com; 5A. Tsyb Medical Radiological Research Center-National Medical Research Center of Radiology, Ministry of Health of Russian Federation, 249031 Obninsk, Russia; alexman6568@gmail.com (A.P.); valerifs@yahoo.com (V.S.); 6The Center for Peace, Hiroshima University, Hiroshima 7300053, Japan; mhoshi@hiroshima-u.ac.jp

**Keywords:** residual radiation, internal radiation exposure, radiation injury, testis, prostatic function, rat

## Abstract

To investigate the biological effects of internal exposure of radioactive ^56^MnO_2_ powder, the major radioisotope dust in the soil after atomic bomb explosions, on male reproductive function, the gene expression of the testes and the prostate was examined. Ten-week-old male Wistar rats were exposed to three doses of radioactive ^56^MnO_2_ powder (41–100 mGy in whole body doses), stable MnO_2_ powder, or external ^60^Co γ-rays (2 Gy). Animals were necropsied on Days 3 and 61 postexposure. The mRNA expressions of testicular marker protein genes and prostatic secretory protein genes were quantified by Q-RT-PCR. On Day 3 postexposure, the testicular gene expressions of steroidogenesis-related enzymes, Cyp17a1 and Hsd3b1, decreased in ^56^MnO_2_-exposed groups. Germ cell-specific Spag4 and Zpbp mRNA levels were also reduced. On postexposure Day 61, the Cyp11a1 gene expression became significantly reduced in the testes in the group exposed to the highest dose of ^56^MnO_2_, while another steroidogenesis-related StAR gene mRNA level reduced in the ^60^Co γ-rays group. There were no differences in Spag4 and Zpbp mRNA levels among groups on Day 61. No histopathological changes were observed in the testes in any group following exposure. Expression in the prostatic protein genes, including CRP1, KS3, and PSP94, significantly decreased in ^56^MnO_2_-exposed groups as well as in the ^60^Co γ-rays group on Day 61 postexposure. These data suggest that the internal exposure to ^56^MnO_2_ powder, at doses of less than 100 mGy, affected the gene expressions in the testis and the prostate, while 2 Gy of external γ-irradiation was less effective.

## 1. Introduction

Notwithstanding the effects of the initial radiation from the atomic bombings in Hiroshima and Nagasaki, there have been concerns regarding the potentially significant influence of the residual radioactive dust on the health of those exposed. The people who moved to these cities a short time after detonation were only exposed to residual radiation, likely via inhaling radioactive dust, and were reported to suffer from acute radiation syndromes [1]. A primary source of residual radiation was ^56^Mn, a radioisotope produced in the soil by the neutron beam from an atomic bomb explosion [2]. We investigated the biological effects of neutron-activated ^56^MnO_2_ powder in Wistar rats to gain a better understanding of the significance of residual radiation [3,4,5]. Our dosimetry data demonstrate that the highest absorbed doses from internal radiation were found in the gastrointestinal tract, skin, and lungs, while the highest whole-body dose was 100 mGy. Interestingly, at these low radiation doses, internal exposure to ^56^MnO_2_ significantly increased the serum alanine aminotransferase (ALT) level [3]. Histopathological changes were also noted in the small intestine and lungs [4].

The male reproductive system is a sensitive target for radiation exposure [6,7,8]. The Chernobyl accident victims exposed to radioactive particles from the reactor (1–5.5 Gy) suffered from the impairment of exocrine and endocrine testicular function [9]. The effects of residual radioactive dust after an atomic bombing on the male function have been considered but are not yet understood well [10]. Therefore, we investigated the effects of ^56^MnO_2_ particles on the male function in our model. The rat spermatogenesis is disturbed in the testes by radiation exposure, although the sensitivities vary depending on the strain of rats, with lower sensitivities in SD and Wistar rats and higher sensitivities in Brown–Norway and Lewis rats [6,11]. As the majority of studies, including ours, use SD and Wistar rats, when examining the toxicological effect of chemicals or radiation, morphological changes in the testes are not a sensitive endpoint unless doses over 5 Gy of external radiation are applied. In our previous studies of rats exposed to ^56^MnO_2_, testicular histology was examined on postexposure Days 3, 14, and 60, without showing any significant changes as well [4]. However, recent studies demonstrated that changes in testicular gene expressions are useful markers to identify the functional changes in the testes due to ionizing radiation or chemicals [12,13,14]. In the present study, mRNA levels were determined by the quantitative RT-PCR to assess the effects of low-dose radiation exposure with ^56^MnO_2_ on the testes. In addition, mRNA expressions of prostatic secretory protein genes were measured in the prostate to further analyze the effects of ^56^MnO_2_ on male reproductive function. Animals were examined only on postexposure Days 3 and 60 in the present study to increase the number of rats in each group for quantitative analysis since the capacity of the ^56^MnO_2_ exposure system was limited.

## 2. Results

### 2.1. Estimated Doses of Internal Irradiation

The estimated accumulated doses of internal irradiation from ^56^MnO_2_ in each organ were described previously [3]. The whole-body doses of internal irradiation were 41 ± 8, 91 ± 3, and 100 ± 10 mGy in Mn56×1, Mn56×2 and Mn56×3 groups, respectively. The higher absorbed doses were found in the colon (90 ± 61, 520 ± 110, and 760 ± 170 mGy) and skin (71 ± 23, 110 ± 2.3, and 140 ± 170 mGy) for the Mn56×1, Mn56×2, and Mn56×3 groups, respectively. The calculated absorbed doses for the testes and the prostate were less than 0.3 mGy (Mn56×1 group), 0.6 mGy (Mn56×2 group), and 1.0 mGy (Mn56×3 group).

### 2.2. Body and Testes Weight and Serum Testosterone Level

Body weight and relative testes weight on Days 3 and 61 after exposure are summarized in Table 1. There were no significant differences in the testes weight on either day. Serum testosterone levels decreased significantly in the Mn56×2 and Co-60 groups on Day 61 postexposure.

### 2.3. Histology of the Testes

Representative histology of the testes with HE staining on Days 3 and 61 postexposure in Mn56×3, Co-60, and the control are shown in Figure 1. No histological changes were found in the testes in any group.

### 2.4. Effects on mRNA Expression Levels of Leydig Cell-Specific Steroidogenesis Related Genes

Relative mRNA expressions of steroidogenesis-related genes, Cyp11a1, Cyp17a, Hsd3b1, and StAR in each group, are presented in Figure 2. On postexposure Day 3, Cyp17a and Hsd3b1 mRNA expressions significantly reduced in the Mn56 groups. On Day 61 postexposure, the Cyp11a1 mRNA level became significantly low in the Mn56×3 group, while the StAR mRNA level reduced in the Co-60 group. None of these mRNA levels were significantly modified in the cold Mn group.

### 2.5. Effects on mRNA Expression Levels of Sertoli Cell and Germ Cell-Specific Genes

Expression levels of Sertoli cell-specific genes, Cld11 and Clu, and germ cell-specific genes, Spag4 and Zpbp, were also determined (Figure 3). No significant changes were noted in Cld11 or Clu, on either Day 3 or 61. The expressions of two germ cell-specific genes, Spag4 and Zpbp, were significantly reduced in Mn56 groups on Day 3 but recovered on Day 61.

### 2.6. Prostatic Secretory Protein mRNA Expressions

The mRNA levels of three major ventral prostate secretory protein genes, Prst C3, CRP1, and KS3, were measured in the ventral prostatic lobe on Day 61 (Table 2). The expression of a gene encoding PSP94, a dorsolateral prostate secretory protein, was also determined. KS3 mRNA levels decreased in Co-60, Mn56×2, and Mn56×3, while CPR1 mRNA levels decreased in Co-60 and Mn56×2. PSP94 gene expression significantly dropped by half from the control value only in the Mn56×3 group.

## 3. Discussion

To understand the biological effects of the residual radioactive particles following an atomic bombing, we examined male Wistar rats exposed to ^56^MnO_2_ powder and found that internal exposure to this radioactive powder had higher biological impacts than external irradiation [3,4]. In the present study, the effects of internal exposure to ^56^MnO_2_ powder on the male reproductive function were investigated by determining the gene expression changes in the testes as well as the prostate. Although testicular radiation doses were less than 110 mGy, the mRNA levels in several steroidogenesis related genes were affected on Days 3 and 61 postexposure as prostatic protein gene expressions were also downregulated on Day 61 postexposure. Our results suggest that exposure to ^56^MnO_2_ powder significantly affected the male reproductive function associated with reduced gene expressions in the testes and the prostate despite the low radiation doses.

Spermatogenesis in the testes is sensitive to both radiation and chemical toxicants [7,8]. The damage to spermatogonia caused by these agents led to a progressive depletion of differentiating germ cells. In rodents, there are dramatic inter-strain differences in the sensitivity of the testes to ionizing radiation [11,15,16]. While irradiation severely blocks spermatogenesis in Brown–Norway and Lewis rats, the testes in SD and Wistar rats are more resistant. The germ cell differentiation can be recovered quickly following irradiation of 2 Gy in Wistar rats and even of 5 Gy in SD rats [6], which was consistent with our present results showing that the testicular histopathology is similar between the control and Co-60 groups. However, even in these strains of rats, testicular gene expression could be affected by exposure to radiation or chemicals, representing the possible adverse effects on the gland. A study analyzing the effects of di(n-byryl) phthalate, an endocrine disruptor, found that the changes in the testicular gene expression were sensitive markers in SD rats [14]. Their data indicate that steroidogenesis-related genes and germ cell marker genes could be useful in evaluating the toxicity of the chemical. Another study in SD rats evaluating the testicular toxicity of Icariin identified the specific changes in mRNA expressions, including steroidogenesis-related genes and Sertoli cell-specific genes in the testes, where no histopathological alterations were found [12]. More systematic profiling of transcriptome in the testes after irradiation was reported in mice locally exposed to 1 Gy of X-radiation [17]. The authors found five clusters of gene expression changes that could be associated with alterations of germ cells and somatic cells. Based on these previous testicular gene expression studies, we chose to determine the expression of steroidogenesis related genes specifically expressed in Leydig cells, Sertoli cell-specific genes, and germ cell-associated genes, including Spag4 and Zpbp, to evaluate the effects of ^56^MnO_2_ exposure. On Day 3 postexposure, a series of gene expression decreased, which may indicate that the functions in both germ and somatic cells were affected by the exposure. Interestingly, external Co-60 γ irradiation only reduced Spag4 and Zpbp gene expressions to a lesser extent, suggesting that the internal exposure to ^56^MnO_2_ had a stronger effect. Internal exposure to ^56^MnO_2_ and external exposure to ^60^Co-γ rays had a differential impact on long-term gene expressions in the testes. On Day 61 postexposure, the Cyp11a1 mRNA level was significantly lower in the Mn56×3 group, while StAR gene expression became significantly reduced in the Co-60 group. A recent study examining the testicular gene expression in Wistar rats also found a similar effect: a marked decrease in StAR gene expression after gamma irradiation [13]. It is interesting that both internal ^56^MnO_2_ exposure and external γ-ray exposure affect the steroidogenesis gene expressions, likely resulting in reduced testosterone production. The functional changes in Leydig cells by ionizing radiation are not likely related to the histological changes in the testes [18,19]. Similarly, the alteration in steroidogenesis related gene expressions regulating the Leydig cell function may not lead to any morphological changes, as shown in this study. Previous investigations suggested that Sertoli cells are more resistant to radiation than Leydig cells in rats [18,19], which correlates with our gene expression results, showing that Cld11 and Clu gene mRNA levels did no changes after exposure, save for a slight reduction in Clu mRNA level in Mn56×3 on Day 3.

To further evaluate the effect of ^56^MnO_2_ on male reproductive function, we examined prostatic activities. The rodent prostate consists of morphologically-separated ventral, dorsolateral, and anterior (coagulating gland) portions, each of which secretes different proteins representing prostatic development and activity [20]. We determined mRNA levels of ventral prostatic secreted proteins, prstC3, CRP1, and KS3, and a dorsolateral prostatic protein, PSP94. These mRNA expressions significantly decreased in the Mn56×3 group in Day 61 postexposure, suggesting that the ^56^MnO_2_ exposure posed the prostatic function, probably via the reduction in serum testosterone levels.

The gene expressions significantly changed in the testes and the prostate in Mn56 groups, despite their low internal radiation doses (less than 1 mGy). In this respect, it should be noted that estimations of internal irradiation of each organ were based upon the measured radioactivity of the organs. However, in the case of the testes, which were anatomically located outside of the body not being retracted inside, as the animals were asleep during the exposure period, they were additionally subjected to irradiation from adjacent skin directly in contact with ^56^MnO_2_ powder. Considering the maximal energy of 2.85 MeV of β rays from ^56^Mn with a range of approximately 4 mm in the tissue, the additional testicular dose should have been 110 mGy in the Mn56×3 group according to the calculation method we previously reported [21]. However, this was not the case in the prostate, an internal organ. The reduction in mRNA levels of prostatic proteins was likely the result of decreased serum testosterone levels.

The body distribution of certain radionuclides following internal exposure depends on their specific chemical nature [22,23] When rats are exposed to stable and insoluble molecules such as MnO_2_, these molecules are delivered to the gastrointestinal tract, skin, and lungs. It is believed that irradiation from radioactive particles may be less hazardous in terms of carcinogenesis than the same activity uniformly distributed [24]. However, radioactive particles could induce stronger biological impacts in “target organs”, as we previously found histological changes in the lung and the small intestine in rats internally exposed to ^56^MnO_2_ powder [24]. The present study again indicated high biological impacts on certain target organs by the radioactive particles. The effects on the testicular gene expression by ^56^MnO_2_ were evidently higher than the effects caused by 2 Gy of external γ irradiation.

Concerns of the effects of ionizing radiation on male reproductive function have been raised as radiation exposure becomes more common in both diagnostic and therapeutic procedures [25]. Human testes are known to be more radiosensitive than those of rodents. Human spermatogenesis could be disturbed following external irradiation as low as 0.3 Gy [26]. Doses greater than 1 Gy significantly reduced the number of spermatogonia and spermatocytes, which require 9–19 months to recover. In cases of victims of the Chernobyl accident, one study examined 12 men receiving doses varying from 1 to 5.5 Gy. It reported that the radiation exposure was associated with impairment of exocrine and endocrine testicular function [9]. Another study investigated men who cleaned the region around the reactor and received doses of 0.16 ± 0.08 Gy; it did not find any disruption in the endocrine status and spermatogenesis. However, this study was conducted 7–9 years following the exposure [27]. People in northeastern Kazakhstan adjacent to the Semipalatinsk nuclear test site had been exposed to radioactive fallout after nuclear tests conducted by the former Soviet Union [28]. One study examined the sex ratio in the offspring of these people as an indicator of the radiation effect on reproductive health and did not find any significant changes [10]. However, this result does not exclude the possibility of adverse effects caused by radiation fallout on the reproductive function in the exposed population, since sex-linked lethal mutations in germ cells, which require high doses of radiation exposure, have been postulated to cause changes in the sex ratio of the offspring [29]. The present study demonstrated that rats exposed to ^56^MnO_2_ powder decreased the gene expressions related to male reproductive function, including steroidogenesis related genes in Leydig cells and prostatic secreted protein genes. Further studies are recommended to determine whether these changes lead to any adverse effect on male reproductive health.

## 4. Materials and Methods

### 4.1. Animals

The animal experiment design was described previously [3]. Ten-week-old male Wistar rats were purchased from Kazakh Scientific Center of Quarantine and Zoonotic Diseases, Almaty, Kazakhstan. Animals were housed in plastic cages (two rats/cage) and maintained with free access to basal diet and tap water. The room was maintained at 19–22 °C with a relative humidity of 30–70% and a 12-h light cycle. The animal facility was of the conventional type, although the purchased rats were specific-pathogen free animals. Rats were randomly divided into 6 groups, adjusting the average body weights to be similar among groups: Mn56×1 group (*n* = 17), Mn56×2 group (*n* = 17), Mn56×3 group (*n* = 17), Co-60 group (*n* = 14), cold Mn group (*n* = 14), and control group (*n* = 14). Mn56×1, Mn56×2, and Mn56×3 groups were exposed to 3 different activities of ^56^MnO_2_ powder (100 mg) of 2.7 × 10^8^, 5.5 × 10^8^, and 8 × 10^8^ Bq, respectively. The cold Mn group was exposed to nonradioactive MnO_2_ powder (100 mg). Three rats from each Mn56 group were necropsied to assess absorbed dose 0.5 h following exposure. The Co-60 group received 2 Gy of external ^60^Co γ-ray whole-body irradiation. From each group, 7 rats were necropsied on postexposure Days 3 and 61 between 11:00 and 17:00. The rats were placed into a box containing isoflurane anesthetic gas (Fujifilm Wako Pure Chemical Co., Tokyo, Japan), and then euthanized by removing and collecting whole blood from an abdominal artery. The testes, the ventral prostate, the dorsolateral prostate, and the seminal vesicles were dissected. Half of the testes were stored in RNA Save solution (Biological Industries Ltd., Beit Alfa, Israel) for RNA extraction, and the other half was fixed in 10% formalin and embedded in paraffin. Sections of 4 μm thickness were prepared and stained with hematoxylin and eosin. Prostate tissue samples were collected only on Day 61 and stored in RNA Save solution. Ethics Committee Approval (document #3-30.11.2018) was received from the Animal Experiment Ethics Committee of Semey Medical University, Semey Kazakhstan.

### 4.2. Irradiation and Dosimetry

Details of irradiation using ^56^MnO_2_ powder and the internal dose estimation have been described previously [5]. Briefly, MnO_2_ powder (Rare Metallic Co., Tokyo, Japan, particle diameters ranging from 1–19 µm) was radio-activated by neutron beam in the Baikal-1 nuclear reactor at the National Nuclear Center, Kurchatov, Kazakhstan. Thermal neutron fluencies of 4 × 10^14^, 8 × 10^14^, and 1.2 × 10^15^ n/cm^2^ were applied to each 100 mg of MnO_2_ powder to produce ^56^MnO_2_ with activities of 2.7 × 10^8^, 5.5 × 10^8^, and 8 × 10^8^ Bq/100 mg (for Mn56×1, Mn56×2, and Mn56×3 groups, respectively). This ^56^MnO_2_ powder was then air-pressure sprayed into sealed exposure boxes containing 8 or 9 rats per box. Following 1 h of exposure, the animals were moved to new cages. Thirty minutes later, three rats per group were euthanized. A section of each organ was dissected, and the radioactivity was measured with a γ-spectrometer. Absorbed fractions of energy from β and γ-irradiation of ^56^Mn in each organ, as well as that for the whole body, were calculated using the Monte Carlo code (version MCNP-4C) and the mathematical phantom of the rat. Absorbed doses of internal radiation were estimated on the basis of measured radioactivity of ^56^Mn in each organ using calculated values of absorbed fractions of energy in these organs. All groups of animals were brought to the ^56^Mn exposure facility. A ^60^Co radiotherapy machine, Teragam K-2 unit (UJP Praha, Praha-Zbraslav, Czech Republic), was used for whole-body γ-ray irradiation of 2 Gy (1.0 Gy/min).

### 4.3. Measurement of mRNA Levels by Quantitative RT-PCR

Total RNA was prepared using Isogen II (Nippon Gene Co., Tokyo, Japan) from sections of testes and prostatic tissue stored in RNA Save solution. First-strand cDNA was synthesized by incubating 3 µg total RNA with 100 U of ReverTra Ace reverse transcriptase (Toyobo Co., Osaka, Japan) with a mixture of 20 pmol random hexamers pdN6 and 5 pmol oligo-dT(15) primers (Takara Bio Inc., Kusatsu, Japan). A quantitative PCR instrument, StepOnePlus (Applied Biosystems/Life Technologies Co., Carlsbad, CA, USA), was employed for measurement of cDNA with a KAPA SYBR Fast qPCR Kit (Kapa Biosystems, Inc., Woburn, MA, USA). Prior to quantitative analysis, the PCR products were prepared separately and purified by gel electrophoresis. The DNA sequences were confirmed by Fasmac Co., Ltd. (Atsugi, Japan). The extracted fragments were used as standards for quantification. PCR conditions were 30 s initial denaturation followed by 40 cycles of 5 s at 95 °C and 35 s at 60 °C. The measured mRNA levels were normalized with reference to the levels of β-actin mRNA. Specific primer sets for testicular genes are listed in Table 3. The Q-PCR primers for prostate secretory proteins were described previously [30].

### 4.4. Measurement of Serum Testosterone

Serum testosterone levels were measured by testosterone ELISA Kit (Cayman Chemical Co., Ann Arbor, MI, USA) following the company’s instructions.

### 4.5. Statistical Analysis

All values are expressed as mean ± standard error of the mean. Dunnett’s test was performed to compare each treated group with the control group.

## 5. Conclusions

^56^Mn is one of the dominant sources of the residual radiation found in the soil after an atomic bomb explosion. We investigated the effect of ^56^MnO_2_ powder on the gene expressions in the testes and the prostate. The mRNA levels in several testicular steroidogenesis related genes were affected postexposure with ^56^MnO_2_ powder despite the low radiation dosages (less than 110 mGy), while external γ-irradiation at the dose of 2 Gy showed little impact on these parameters. Our results suggest that exposure to ^56^MnO_2_ powder had a significantly higher biological impact than the external irradiation on the male reproductive function associated with the examined gene expressions.

## Figures and Tables

**Figure 1 ijms-21-04989-f001:**
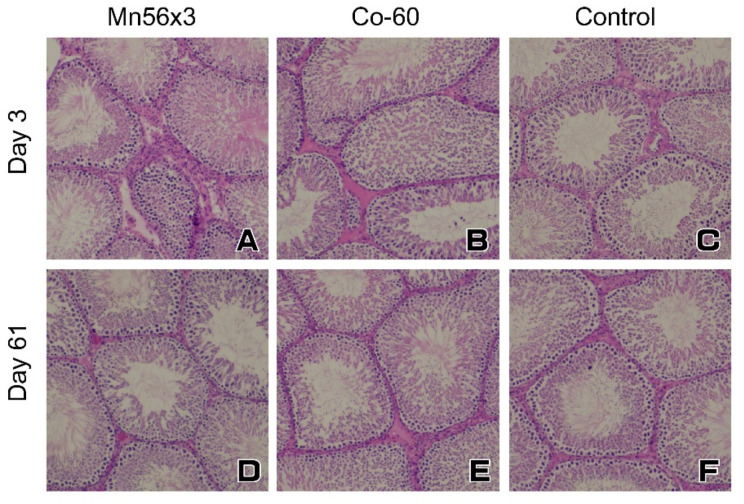
Testes of rats on Day 3 (**A**–**C**) and Day 61 (**D**–**F**) following ^56^MnO_2_ powder or ^60^Co-γ exposure. There were no significant histological alternations in the testes among the groups: Mn56×3 (**A**,**D**); Co-60 (**B**,**E**); and the control (**C**,**F**). HE staining, original magnification 20×.

**Figure 2 ijms-21-04989-f002:**
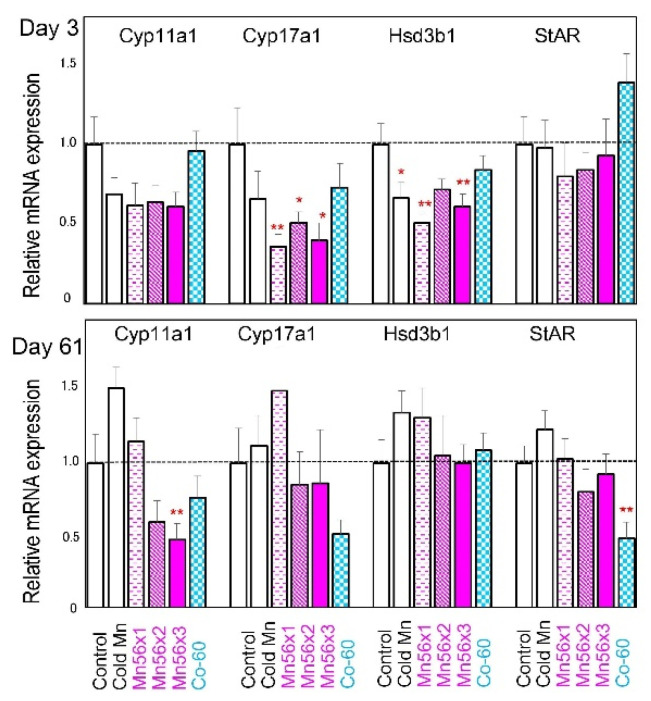
Relative mRNA expression levels of Cyp11a1, Cy17a1, Hsd3b1 and StAR genes in the testes of the rats on Day 3 (**top**) and Day 61 (**bottom**) after the exposure to ^56^MnO_2_ powder (Mn56×1, Mn56×2, and Mn56×3), Cold MnO_2_ powder (Cold Mn), or ^60^Co-γ exposure (Co-60). * *p* < 0.05 and ** *p* < 0.01 vs. control.

**Figure 3 ijms-21-04989-f003:**
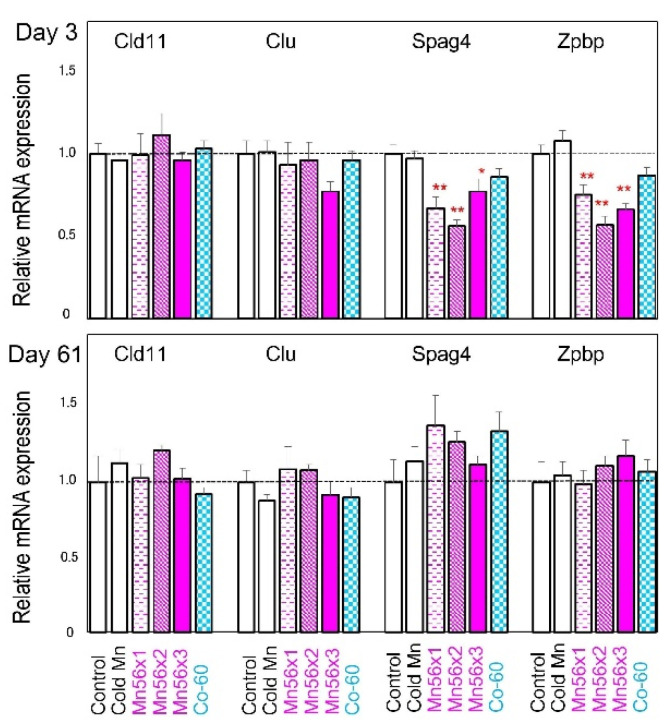
Relative mRNA expression levels of Cld11, Clu, Spag4, and Zpbp genes in the testes of the rats on Day 3 (**top**) and Day 61 (**bottom**) following exposure to ^56^MnO_2_ powder (Mn56×1, Mn56×2, and Mn56×3), Cold MnO_2_ powder (Cold Mn), or ^60^Co-γ exposure (Co-60). * *p* < 0.05 and ** *p* < 0.01 vs. control.

**Table 1 ijms-21-04989-t001:** Body and testis weights and serum testosterone in rats exposed to ^56^MnO_2_, ^60^Co γ-rays, and cold MnO_2_.

Groups		Body Weight (g)	Testes (g/kg bw)	Testosterone (pg/mL)
Day 3	Control	248 ± 16	10.5 ± 1.1	1.2 ± 0.30
	Cold Mn	235 ± 14	11.4 ± 0.7	0.94 ± 0.13
	Mn56×1	235 ± 11	11.7 ± 0.6	0.7 ± 0.24
	Mn56×2	245 ± 16	11.3 ± 0.6	0.8 ± 0.17
	Mn56×3	237 ± 12	12.2 ± 0.6	not determined
	Co-60	234 ± 14	11.5 ± 0.8	1.15 ± 0.33
Day 61	Control	330 ± 17	9.3 ± 0.7	1.45 ± 0.35
	Cold Mn	337 ± 19	9.8 ± 0.4	1.3 ± 0.14
	Mn56×1	371 ± 21	9.2 ± 0.6	1.42 ± 0.14
	Mn56×2	337 ± 17	9.1 ± 0.5	0.68 ± 0.26 *
	Mn56×3	353 ± 17	9.1 ± 0.5	0.75 ± 0.23
	Co-60	328 ± 23	9.4 ± 0.5	0.59 ± 0.11 *

Each value shows mean ± SEM (*n* = 6 or 7, each group); * indicates significantly different from control by *p* < 0.05.

**Table 2 ijms-21-04989-t002:** mRNA levels of secretory protein genes in the prostate.

Groups	PrstC3 (fmol/fmol βact)	CRP1 (fmol/fmol βact)	KS3 (fmol/fmol βact)	PSP94 (fmol/fmol βact)
Control	651 ± 45	158 ± 22	20 ± 2.6	103 ± 7.3
Cold Mn	566 ± 38	168 ± 26	13.5 ± 2.2	83 ± 11.2
Mn56 × 1	716 ± 82	206 ± 34	18.3 ± 2.9	86 ± 33.8
Mn56 × 2	611 ± 103	77 ± 14 *	8.3 ± 1.6 **	75 ± 9.0
Mn56 × 3	453 ± 24	91 ± 10	9.8 ± 2.8 *	49 ± 10.7 *
Co-60	557 ± 79	79 ± 19 *	11.8 ± 2.6 *	71 ± 9.4

Prostate tissue samples were collected and examined only on Day 61 postexposure. Each value shows mean ± SEM (*n* = 6 or 7, each group). * *p* <0.05 and ** *p* < 0.01 significantly different from each control.

**Table 3 ijms-21-04989-t003:** Q-PCR primers.

Gene	GenBank Accession#	Q-PCR Primer Sequences (5′–3′)	
		Forward	Reverse
Cyp11a1	NM_017286	TCCTCCCTGGTTACGTGCAG	GCAGAATAAGGAGCACCCCAG
Cyp17a1	NM_012753	CAGCCAGATCAGTTCATGCCT	GACAAAGAGCTCCTGACGGG
Hsd3b1	NM_001007719	AGAGAGATCTGGGCTATGTGCC	ACACCCAGAACCACATCCTTG
StAR	NM_031558	ACCTGCATGGTGCTTCATCC	GCTGGCGAACTCTATCTGGGT
Cld11	NM_053457	TCCTCCCTGGTTACGTGCAG	GCAGAATAAGGAGCACCCCAG
Clu	NM_053021	TGCTTCATTCCCTCCAGTCC	TGGGTTGTCACTGTGGAGACC
Spag4	NM_031792	CCAAGCTGATGATGACGAGACT	GGCCCCAGTTGCTTAAAATCT
Zpbp	NM_001025139	TTCAGCAAGTGGAAGTCCTGG	ACACAGCACTCAGGACACTTGG

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
