# Peer review of "Low-Dose Radiation Exposure with 56MnO2 Powder Changes Gene Expressions in the Testes and the Prostate in Rats"

_ijms, 2020, doi:10.3390/ijms21144989_

Round 1
Reviewer 1 Report
First of all, I want to congratulate you on the clear and easy to understand work you have done. In general the data is very well expressed and makes understanding very easy.
I only have a series of comments to make, especially from a formal point of view.
In general, English is fine, although it needs some small corrections, such as:
- On line 61 it says "Lewis of rats" and it must say "Lewis rats".
- On line 80 instead of putting “weight of the testes” you should put “testes weight.
I recommend that you review the text to correct minor grammatical errors that may have arisen.
In line 52 you talk about "serum ALT" but you never mention the meaning of ALT. It would be recommended that they write “alanine aminotransferase (ALT)” so that the meaning of ALT is clear.
In the figures, the word “fig.1, fig.2…” appears at the top right. I do not think they need to appear as it is well indicated in the figure caption.
In figure 1, I recommend that you highlight the letters A, B... better because they are not very legible.
On line 118 you refer to Table 3 when the data you describe refers to Table 2. Please correct that error.
As for questions that have arisen when reading your article I have the following questions:
Why have you used two such separate study time intervals, 3 and 61 days? Would it have been more representative of the changes that could occur if an intermediate group were included after, for example, 30 days?
I find it striking that there are no significant differences between the control group and the Mn56x3 group on day 61 (Table 1) in terms of Testosterone concentration. Could you explain the reason for the non-significance by looking at the results there are? Is it due to high data dispersion?
When the prostate results are shown (lines 117-121), I notice that you only put the results on day 61, why aren't the results on day 3? Is it because there are no significant variations or because the study has not been done?
In line 120 you comment that there is a significant decrease in CRP1 in Co-60, Mn56x2 and Mn56x3 but in the table there are no significant differences in Mn56x3. Please, if it is an error, correct it where it should (or the table or the text).
Finally, a question that has arisen to me is, if there are no histological changes in the study groups but changes in the expression levels of the genes studied, have you been able to study at the cellular level if changes in the expression of proapoptotic or anti apoptotic proteins are observed or if there are changes in cell proliferation rates? Are there changes in the seminal parameters of these rats? And in general, have they observed any histopathological relationship with the observed genes alterations?
Author Response
Responses to Reviewer#1’s comments:
Thank you very much for your kind words and constructive comments.
1.In general, English is fine, although it needs some small corrections, such as:
- On line 61 it says "Lewis of rats" and it must say "Lewis rats".
- On line 80 instead of putting “weight of the testes” you should put “testes weight.
I recommend that you review the text to correct minor grammatical errors that may have arisen.
I reviewed the text again for correcting errors.
Re1:
In line 52 you talk about "serum ALT" but you never mention the meaning of ALT. It would be recommended that they write “alanine aminotransferase (ALT)” so that the meaning of ALT is clear.
It was spelled out as you recommended (line 52)
2.In the figures, the word “fig.1, fig.2…” appears at the top right. I do not think they need to appear as it is well indicated in the figure caption.
In figure 1, I recommend that you highlight the letters A, B... better because they are not very legible.
Re2:
Labels “fig.1, fig.2…” were removed from figures. Letters in photos of Figure 1 were highlighted. In Figures 2 and 3, black and white graphs were replaced with color graphs for clarity. (Figures 1-3)
3.On line 118 you refer to Table 3 when the data you describe refers to Table 2. Please correct that error.
Re3:
Corrected (line 120)
4.As for questions that have arisen when reading your article I have the following questions:
Why have you used two such separate study time intervals, 3 and 61 days? Would it have been more representative of the changes that could occur if an intermediate group were included after, for example, 30 days?
Re4:
In our previous study, we examine the histology in major organs on postexposure days 3, 14, and 60 and found no pathological changes in the testes on either day, although each group consisted of only 3rats. In the present study, we intended to increase the number of animals/ group so that we could get data fit for quantitative analysis. But due to the limitation in the capacity of our radiation exposure facility (and our budget), we need to limit the time points. We thought a (relatively) long-term effects (if any) would be more important for examining the testes and the prostate related to male reproductive health. Regarding this, Introduction was extended (line 63-65, 71-73)
5.I find it striking that there are no significant differences between the control group and the Mn56x3 group on day 61 (Table 1) in terms of Testosterone concentration. Could you explain the reason for the non-significance by looking at the results there are? Is it due to high data dispersion?
Re5:
Yes, the serum T levels showed relatively large variations as the S.E. indicated. Mn56x3 consisted of 6 rats instead of 7, which also counts for statistical test results. Dunnett’s test showed that p vale of Mn56x3 against control was 0.088, while that of Mn56x2 was 0.037.
6.When the prostate results are shown (lines 117-121), I notice that you only put the results on day 61, why aren't the results on day 3? Is it because there are no significant variations or because the study has not been done?
Re6:
Regrettably, we did not collect prostate tissue samples on day 3. Regarding the effects of 56Mn on the male function, we initially planned to examine only the testes. The idea of examining the prostate gene expressions came up after we finished the necropsy on day 3. The statement that prostate tissues were stored only on day 61 was added to the footnote of Table2.
7.In line 120 you comment that there is a significant decrease in CRP1 in Co-60, Mn56x2 and Mn56x3 but in the table there are no significant differences in Mn56x3. Please, if it is an error, correct it where it should (or the table or the text).
Re7:
Reduction in CRP1 mRNA was not significant in Mn56x3 as Table 2 showed. The text was corrected (line 121-122). Thank you.
8.Finally, a question that has arisen to me is, if there are no histological changes in the study groups but changes in the expression levels of the genes studied, have you been able to study at the cellular level if changes in the expression of proapoptotic or anti apoptotic proteins are observed or if there are changes in cell proliferation rates? Are there changes in the seminal parameters of these rats? And in general, have they observed any histopathological relationship with the observed genes alterations?
Re8:
An early decrease in mRNAs of germ cell associated genes, Spag4 and Zpbp might indicate a reduction in germ cell population as Ref.#17 suggested. That may be the result of a temporal decrease in cell proliferation without significantly altering the morphology. But we have no data. On the other hand, changes in mRNAs of steroid genesis related genes may not produce any morphological changes in the testis, since previous studies (Refs#18 & 19) demonstrated that radiation exposure affected the steroidogenesis in the Leydig cell but not morphologically. Discussion was extended regarding this (line 169-172).
Reviewer 2 Report
The manuscript by Fujimoto et al. describes the effects of the internal exposure to 56MnO2 powder on the gene expression of the testes and the prostate in Wistar rats.
The present work represents an addition to previous published studies by the same authors.
The whole study on the effects of residual radioactive dust is very interesting.
I have some questions:
- why did you select days 3 and 61 as time points? what about intermediate times (e.g.day 14)?
- why did you measure the levels of the three major ventral prostate secretory protein genes only on day 61 (table 2)? what about the day 3?
Author Response
Responses to Reviewer#2’s comments.
Thank you very much for your kind words and constructive comments.
The manuscript by Fujimoto et al. describes the effects of the internal exposure to 56MnO2 powder on the gene expression of the testes and the prostate in Wistar rats.
The present work represents an addition to previous published studies by the same authors.
The whole study on the effects of residual radioactive dust is very interesting.
I have some questions:
1.why did you select days 3 and 61 as time points? what about intermediate times (e.g.day 14)?
Re1:
In our previous study, we examine the histology in major organs on postexposure days 3, 14, and 60 and found no pathological changes in the testes on either day, although each group consisted of only 3rats. In the present study, we intended to increase the number of animals/ group so that we could get data fit for quantitative analysis. But due to the limitation in the capacity of our radiation exposure facility (and our budget), we need to limit the time points. We thought a (relatively) long-term effects (if any) would be more important for examining the testes and the prostate related to male reproductive health. Regarding this, Introduction was extended (line 63-65, 71-73)
2.why did you measure the levels of the three major ventral prostate secretory protein genes only on day 61 (table 2)? what about the day 3?
Re2:
Regrettably, we did not collect prostate tissue samples on day 3. Regarding the effects of 56Mn on the male function, we initially planned to examine only the testes. The idea of examining the prostate gene expressions came up after we finished the necropsy on day 3. The statement that prostate tissues were stored only on day 61 was added to the footnote of Table2.